

# WRF-PDAF v1.0: Implementation and Application of an Online Localized Ensemble Data Assimilation Framework

Changliang Shao[1,2], Lars Nerger[2]

[1]China Meteorological Administration Meteorological Observation Centre, Beijing, 100081, China

[2]Alfred-Wegener-Institut, Helmholtz-Zentrum für Polar- und Meeresforschung (AWI), Bremerhaven, 27570, Germany

*Correspondence to*: Changliang Shao (shaocl@cma.gov.cn)

**Abstract.** Data assimilation is a common technique employed to estimate the state and its associated uncertainties in numerical models. Ensemble-based methods are a prevalent choice, although they can be computationally expensive due to the required ensemble integrations. In this study, we enhance the capabilities of Weather Research and Forecasting–Advanced Research

WRF (WRF-ARW) model by coupling it with the Parallel Data Assimilation Framework (PDAF) in a fully online mode. Through minimal modifications to the WRF-ARW model code, we have developed an efficient data assimilation system. This system leverages parallelization and in-memory data transfers between the model and data assimilation processes, greatly reducing the need for file I/O and model restarts during assimilation. We detail the necessary program modifications in this study. One advantage of the resulting assimilation system is a clear separation of concerns between data assimilation method

development and model application resulting from PDAF's model-agnostic structure. To evaluate the assimilation system, we conduct a twin experiment simulating an idealized tropical cyclone. Cycled data assimilation experiments focus on the impact of temperature profiles. The assimilation not only significantly enhances temperature field accuracy but also improves the initial U and V fields. The assimilation process introduces only minimal overhead in run time when compared to the model without data assimilation and exhibits excellent parallel performance. Consequently, the online WRF-PDAF system emerges

as an efficient framework for implementing high-resolution mesoscale forecasting and reanalysis.

## 1 Introduction

Data assimilation (DA) plays a pivotal role in enhancing the precision and dependability of numerical weather prediction (NWP) models, effectively bridging the divide between model simulations and real-world observations. It bolsters the accuracy, proficiency, and trustworthiness of weather forecasts, supplying invaluable insights for a diverse array of applications,

encompassing weather prediction, climate research, and environmental assessments (Lorenc, 1986; Song et al., 2022).

Based on the mode of data transfer between the numerical model and assimilation algorithm, ensemble-based DA computational setups can be categorized into two coupling modes: offline and online DA. In offline DA, data exchanges between the model ensemble and assimilation algorithm happen through disk files. Examples of this approach encompass the Advanced Regional Prediction System Data Assimilation System (ARPSDAS; Xue et al., 2000), the Data Assimilation



Research Testbed (DART; Anderson et al., 2009), the Gridpoint Statistical Interpolation (GSI) Ensemble Kalman filter (EnKF) system (Kleist et al., 2009), the Weather Research and Forecasting model's Community Data Assimilation system (WRFDA; Barker et al., 2012) and WRF-EDAS (Ensemble Data Assimilation System; Zupanski et al., 2011). Offline DA offers convenience for implementing DA procedures in relatively short timeframes. Although the actual I/O time may not be substantial, as e.g., described by Karspeck et al. (2018), offline DA systems incur costs associated with restarting the model

after each analysis cycle for ensemble simulations and the potential redistribution of data.

Online DA is typically implemented by coupling a numerical model and DA algorithm into a single executable program and exchanging data between the model and assimilation in memory. Notable online DA systems include the ensemble DA system by Zhang et al. (2007) based on the Geophysical Fluid Dynamics Laboratory coupled climate model (CM2), and the ensemble DA system (Sun et al., 2022) based on the Community Earth System Model (CESM). Further, the Parallel Data Assimilation

Framework (PDAF, Nerger and Hiller, 2013) provides online DA, e.g., in its implementation with the fully coupled Alfred Wegener Institute Climate Model (AWI-CM; Nerger et al., 2020; Mu et al., 2023). In this paper, PDAF Version 2.0 (http://pdaf.awi.de, last access: February 21, 2023) is adopted to carry out the coupling work. In this version, the interface for observations, named Observation Module Infrastructure (OMI) is completely newly developed.

The WRF-ARW model (Skamarock et al., 2021) has gained extensive usage in regional research and real-time forecasting. It

is a reginal modeling system serving atmospheric research and operational weather prediction communities. Different studies have explored extensive DA works with WRF, such as WRFDA (Wang et al., 2008; Liu et al., 2020), WRF-DART (Kurzrock et al., 2019; Risanto et al., 2021) and WRF-GSI (Yang et al., 2015; Liu et al., 2018). These studies are predominantly grounded in offline DA frameworks, necessitating read and write operations for restart files at each assimilation time step and model restarts for each subsequent forecast phase. This time-intensive approach poses challenges for generating efficient high-

resolution reanalysis. In pursuit of an efficient reanalysis, particularly with the goal of high-resolution WRF-ARW DA, an online DA system coupled with WRF-ARW has emerged as an imperative need. This study presents an extension of WRF-ARW's capabilities by introducing the online-coupled WRF-PDAF (Shao, 2023) DA system, to bolster its potential for mesoscale research and high-resolution DA applications. WRF-PDAF facilitates in-memory data transfer, avoiding the need for repeated model restarts and thus enabling efficient support for high-resolution simulations. Further WRF-PDAF utilized

ensemble parallelization to ensure computational efficiency.

For the application of DA, temperature (T) profile observations have gained significant attention in recent years due to their potential for enhancing atmospheric models and weather forecasts. These observations can be derived from various remote sensing instruments, including radiosondes, dropsonde, ground-based and space-based lidars, and microwave radiometers. The assimilation of temperature profiles into atmospheric models using techniques like Ensemble Kalman Filtering (EnKF) has

yielded substantial improvements in model accuracy and performance (Raju et al., 2014), particularly in the realm of short-term forecasts (Rakesh et al., 2009). Such assimilation aids in capturing mesoscale weather phenomena like convective systems, thunderstorms, and localized rainfall patterns. It contributes to the more faithful representation of atmospheric processes, enhancing the skill of weather forecasts, particularly in regions where traditional observations are sparse or limited (Feng and



Pu, 2023). Assimilating profiles facilitates a more precise vertical profiling of atmospheric parameters, critical for

comprehending the vertical structure of the atmosphere (Holbach, 2023). Real-time assimilation of profiles enables timely updates to atmospheric models, leading to improved nowcasting and short-term forecasts (Pan, 2023). This real-time assimilation empowers models to capture swiftly evolving atmospheric conditions, providing crucial insights for severe weather events and rapid weather developments (Pena, 2023).

This paper serves as an introduction of the fully online-coupled WRF-PDAF system with focus on its development and design.

Additionally, the study assesses the DA behavior of the system in the case of the assimilation of T profiles using the Ensemble Kalman Filtering (EnKF) technique. Twin experiments employing synthetic observations are conducted, and the assimilation results are analyzed, with a specific focus on fully online assimilation of T profiles.

The subsequent sections of this study are organized as follows: Section 2 provides an overview of the WRF model and its configuration, the tropical cyclone case and the twin experiment, and the ensemble filtering algorithm. Section 3 details the

implementation of the ensemble-based online WRF-PDAF DA system, including descriptions of the PDAF system (Sect. 3.1), the augmentation of WRF for DA with PDAF (Sect. 3.2), and discussions on interfaces for model fields and observations (Sect. 3.3). Section 4 encompasses the assessment of the scalability and an evaluation of the assimilation behavior with the online WRF-PDAF system. Finally, Section 5 offers a summary and discusses the findings.

## 2 Methodology

In this section, we introduce the WRF model and its configuration, the tropical cyclone case, the Local Ensemble Square Root Transform Kalman Filter (LESTKF) assimilation scheme, and the twin experiment.

### 2.1 WRF

The WRF model stands as a widely embraced numerical weather prediction system, offering a versatile platform for simulating a broad spectrum of atmospheric processes. Its applicability spans both regional and global weather simulations, all thanks to

its modular structure, enabling tailoring to specific research goals or operational forecasting requirements. The dependability and flexibility inherent in WRF make it an invaluable tool for our study. In this work, we harnessed WRF-ARW version 4.4.1. In this study, we have adopted the idealized tropical cyclone case provided by WRF as our test case. Tropical cyclones represent formidable and destructive meteorological phenomena originating over warm ocean waters near the equator. These potent storms draw their energy from the latent heat release accompanying the ascent and condensation of moist air into clouds

and precipitation. The idealized tropical cyclone case offers a controlled environment for conducting identical twin experiments, evaluating system scalability, and assessing the behavior of DA with WRF-PDAF. The simulation domain encompasses 3000 km x 3000 km x 25 km, comprising 200×200×20 grid points with a horizontal grid spacing of 15 km and a vertical grid spacing of 1.25 km. The simulation spans a period of six days, commencing on September 1 at 00:00 UTC (010000) and concluding on September 7 at 00:00 UTC (070000). The model employs a time step of 60 seconds and the




Kessler microphysics scheme and the YSU (Yonsei University) boundary-layer physics, with radiation schemes omitted. The initialization of the simulation necessitates both initial and boundary conditions. The initial state establishes a horizontally homogeneous environment defined via the default file employing the Jordan mean hurricane sounding, named "input_sounding" within the WRF directory. The initial state is characterized by immobility (u=v=0) and horizontal homogeneity, with the addition of an analytical axisymmetric vortex in hydrostatic and gradient-wind equilibrium. Additionally, periodic lateral

boundary conditions are imposed to facilitate the simulation process.

## 2.2 LESTKF

The EnKF technique serves as a data assimilation method, amalgamating information from a model state ensemble and observational data to refine the model state variables. EnKFs use an initial state ensemble created by introducing perturbations to the model initial conditions. Subsequently, assimilation updates are performed by estimating analysis increments, taking

into account both the ensemble spread and the misfit between observations and model predictions. In this context, ensemble spread, quantified as the ensemble standard deviation (STD), characterizes the dispersion of the ensemble members around the ensemble mean. The analysis increments derived from this process are then applied to the ensemble members, resulting in updated state variables. The EnKF encompassed various variants suitable for assimilating T profiles due to their capacity to manage the nonlinear dynamics typical of atmospheric models. Examples of such variants include the Local Ensemble

Transform Kalman Filter (Hunt et al. 2007; LETKF) and the Local Error-Subspace Transform Kalman Filter (Nerger et al. 2012; LESTKF).

The LESTKF has found application across diverse studies, encompassing the assimilation of satellite data into atmosphere models (Mingari et al., 2022), ocean models (Goodliff et al., 2019), atmosphere-ocean coupled models (Nerger et al., 2020; Zheng et al., 2020) and hydrological models (Li et al., 2023b). In the context of the LESTKF, the EnKF procedure is efficiently

formulated, facilitating discussion on the unique aspects of DA with respect to the ensemble filter. In the mathematical framework, each state vector comprises model fields transformed into a 1-dimensional vector, represented as $x^f$. The columns of the forecast ensemble matrix $X^f$ hold the $N_e$ state vectors. The analysis equations (1)-(4) facilitate the transformation of the forecast ensemble $X^f$ of $N_e$ model states into the analysis ensemble $X^a$:

$$X^a = X^f\left(w 1_{N_e}^T + \widetilde{W}\right) + \bar{x}^f 1_{N_e}^T \tag{1}$$

$$w = TA(HX^fT)^T R^{-1}(y - H\bar{x}^f) \tag{2}$$

$$\widetilde{W} = \sqrt{N_e - 1} \, TA^{1/2}T^T \tag{3}$$

$$A^{-1} = \alpha(N_e - 1)I + (HX^fT)^T R^{-1}HX^fT \tag{4}$$

Here, $\bar{x}^f$ represents ensemble mean state of the forecast, and $1_{N_e}^T$ is the transpose of a vector of size $N_e$, containing the value one in all elements. The vector $w$ with a size of $N_e$, facilitates the transformation of the ensemble mean from the forecast to

the analysis, while the matrix $\widetilde{W}$ (size $N_e \times N_e$), manages the transformation of ensemble perturbations. The matrix $T$ defined by Eq. (5) projects into the error subspace. $H$ is the observation operator. $R$ is the observation error covariance matrix. $A$ is a





transform matrix in the error subspace. $\alpha$ is the forgetting factor (Pham et al., 1998) used to inflate the ensemble to avoid underestimation of the forecast uncertainty. It leads to an inflation of the ensemble variance by $1/\alpha$.

The forecast ensemble represents an error subspace of dimension $N_e - 1$, and the ensemble transformation matrix and vector are computed in this subspace. Practically, one computes an error-subspace matrix by $L = X^f T$, where $T$ is a matrix with $j = N_e$ rows and $i = N_e - 1$ columns that is defined by

$$
T_{j,i} = \begin{cases} 1 - \dfrac{1}{N_e} \dfrac{1}{\frac{1}{\sqrt{N_e}} + 1} & for\ i = j, j < N_e \\[3ex] -\dfrac{1}{N_e} \dfrac{1}{\frac{1}{\sqrt{N_e}} + 1} & for\ i \neq j, j < N_e \\[3ex] -\dfrac{1}{\sqrt{N_e}} & for\ i = N_e \end{cases} \tag{5}
$$

The matrix $A^{1/2}$ in Eq. (3) is computed using the eigenvalue decomposition of $A^{-1}$, calculated as

$$
USU^T = A^{-1} \tag{6}
$$

where $U$ and $S$ denote the matrices of eigenvectors and eigenvalues. Consequently, $A$ in Eq. (2) is computed as

$$
A = US^{-1}U^T. \tag{7}
$$

Similarly, the symmetric square root $A^{1/2}$ in Eq. (3) is computed as

$$
A^{1/2} = US^{-1/2}U^T. \tag{8}
$$

Each grid point in the model is independently updated through a local analysis step. Only observations falling within specified horizontal and vertical localization radii are considered during grid point updates. Therefore, the observation operator is localized and computes an observation vector within these localization radii. Furthermore, each observation is weighted based on its distance to the grid point (Hunt et al., 2007), and a fifth-order polynomial function with a Gaussian-like shape, following the Gaspari and Cohn (1999) approach, is employed to determine these weights. The localization weights are applied to modify the matrix $R^{-1}$ in Eqs. (2) and (4). As a result, the localization process yields individual transformation weights $w$ and $\widetilde{W}$ for each local analysis domain.

### 2.3 DA twin experiments

Table 1 provides an overview of the experiments conducted in this study. The setups for the control state (Exp. 1, 'True'), the true state (Exp. 2, 'CTRL'), and the free ensemble run (Exp. 3, 'ENS') are consistent with Shao and Nerger (2023). Hourly synthetic observations of T profiles are generated from the true state on a span of 30 hours, starting from 040800 and ending at 051400. Assimilation experiments (Exp. 4-14, 'ANA(0-10)') are carried out based on the ensemble run, using observations from T profiles over 30 analysis cycles. These experiments vary in terms of horizontal localization radii, ranging from 0 to 10 times the horizontal grid spacing (dx), where dx is 15km. The vertical localization radii are identical, matching the height of





the model top. The impact of assimilating T profile observations on the model representation of T, as well as the horizontal
velocities U, V, is assessed by comparing the assimilated states with the true states. These experiments allow us to evaluate
the performance and effectiveness of WRF-PDAF in assimilating observations and improving the model representation of
atmospheric variables.

**Table 1: The design for varying the localization radius (dx=15km)**

| Exp. | Name | Member(s) | DA-Cycle(s) | Localization Radius(km) |
|------|------|-----------|-------------|-------------------------|
| 1 | True | 1 | - | - |
| 2 | CTRL | 1 | - | - |
| 3 | ENS | 40 | - | - |
| 4-14 | ANA(0-10) | 40 | 30 | 0-10dx |


In these twin experiments, synthetic observations are generated directly at the model grid points so that no interpolations are
required. Thus, the observation operator for profile data simply selects the T values at the model grid points. Gaussian noise,
with a standard deviation of 1.2K following Li et al. (2023) is added to the T field of the True run to generate the observations.
Each profile represents a single vertical column of observations located at grid points. These profile data are then assimilated
into the WRF model using the LESTKF. The twin experiments commence at 031200, undergoing a spin-up period of 20 hours.
Following this, observations are assimilated hourly during the analysis period, spanning from 040800 to 051400. Subsequently,
an ensemble forecast is executed without additional assimilation from 051400 until 070000. To apply ensemble inflation, a
forgetting factor $\alpha$, where $0 < \alpha \leq 1$, is employed. In this study, an adaptive scheme for the forgetting factor is adopted,
utilizing the statistical consistency measures outlined by Desroziers, et al. (2005), analogous to Brankart et al. (2003).

## 3 Setup of the Data Assimilation Program


The process of coupling the WRF with the PDAF involves integrating function calls from PDAF into the WRF model code to
enable data assimilation capabilities. This section provides an overview of the assimilation framework and the setup of the DA
program. Firstly, a summary of the PDAF is presented in Section 3.1. The modifications made to enable online coupling are
explained in Section 3.2. Furthermore, Section 3.3 discusses the implementation of the interfaces for model fields and
observation.

### 3.1 Subsection (as Heading 2)

PDAF is an open-source software designed to simplify the implementation and application of ensemble and variational DA
methods. It provides a modular and generic framework, including fully implemented and parallelized ensemble filter
algorithms like LETKF, the LESTKF, the NETF (Tödter and Ahrens, 2015), and the LKNETF (Nerger, 2022), along with



related smoothers and variational methods like 3DVAR or 3DEnVAR following Bannister (2017). PDAF also handles model parallelization for parallel ensemble forecasts and manages the communication between the model and DA codes. Written in Fortran, PDAF is parallelized using the Message Passing Interface (MPI) standard (Gropp et al., 1994) and OpenMP (Chandra et al., 2001; OpenMP, 2008), ensuring compatibility with geoscientific simulation models. However, PDAF can still be used with models implemented in other programming languages such as C and Python.

The filter methods within PDAF are model-agnostic and exclusively operate on abstract state vectors, as detailed in in Sect. 2.3 for LESTKF. This design promotes the development of DA techniques independently from the underlying model and simplifies the transition between different assimilation approaches. Model-specific tasks, such as those concerning model fields, the model grid, or assimilated observations, are executed through user-provided program routines based on existing template routines. These routines are equipped with specified interfaces and are invoked by PDAF as call-back routines. Thus,

the model code executes PDAF routines, which in turn call the user routines. To streamline these interactions, calls to PDAF are integrated into interface routines. These routines define the parameters for invoking the PDAF library routines before the actual PDAF routine is executed. Consequently, this approach minimizes changes required within the model code itself, as it mandates only a single-line call to each interface routine – a total of three routines. This call structure presents the advantage of enabling the call-back routines to exist within the context of the model, thus allowing them to be implemented in a manner

akin to model routines. Additionally, the call-back routines can access static arrays allocated by the model, such as through Fortran modules or C header files. This capability facilitates the retrieval of arrays storing, e.g., model fields or grid information, exemplifying the versatility of the system.

## 3.2 Augmenting WRF for DA with PDAF

We adopt a fully online coupling strategy for DA here. This approach assumes the availability of an adequate number of

processes to support concurrent time stepping of all ensemble states, thereby simplifying the implementation. Each ensemble state is integrated by one model task, which can encompass several processes to e.g. allow for domain decomposition. This approach allows each model task to consistently progress forward in time. While the general strategy for online coupling of DA remains consistent with prior studies (Nerger and Hiller, 2013, Nerger et al., 2020; Mu et al., 2023), we present a comprehensive description here to illustrate the implementation of the coupling process for the WRF model. The augmentation

of WRF with DA functionality can be visualized as depicted in Fig. 1.

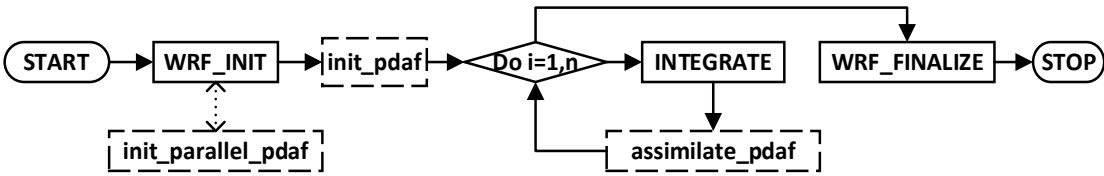



**Figure 1: General program flow of WRF-PDAF. Solid boxes indicate routines in WRF that require parallelization adjustments for data assimilation. Dashed boxes represent essential additions to the model code. Solid lines represent flow, while the dotted line marks a function call inside a routine.**

In Fig. 1, solid boxes delineate the typical flow of the WRF model flow. The program initiates in WRF_INIT, initializing

parallelization, followed by activating all relevant processes. Subsequently, the model is initialized, incorporating grid configuration and initial fields retrieval from files. After completing model initialization, the time-stepping process commences in the routine INTEGRATE. Following time stepping, WRF undergoes cleanup in WRF_FINALIZE, finalizing parallelization, and concluding the program.

Dashed boxes signify essential additions to the model code for online-coupling with PDAF. These additions involve subroutine

calls that serve as interfaces between the model code and the DA framework. By incorporating these subroutine calls, the DA functionality seamlessly integrates into the WRF code, allowing WRF to utilize the DA algorithms. Typically, these subroutine calls entail single-line additions and can be enclosed in preprocessor checks to enable users to activate or deactivate the data assimilation extension during compilation.

The general functionality of the inserted routines is analogous to their roles in a coupled model system (Nerger et al., 2020) as

follows:

**Init_parallel_pdaf**: This routine is merged into the initialization phase to commence parallelization and modify the model parallelization for running an ensemble of model tasks. The parallelization of WRF adheres to the MPI standard. It is initialized at the outset of the program, generating the MPI_COMM_WORLD communicator that encompasses all program processes. Domain decomposition is employed, with each process computing a designated region within the global domain. For ensemble

DA, init_parallel_pdaf adapts the parallelization to accommodate the concurrent computation of multiple model tasks. Achieving this entails partitioning the MPI_COMM_WORLD communicator into communicators for WRF model tasks, termed COMM_model, each with the same number of processes as used by the original domain decomposition. Each communicator within COMM_model represents a distinct model task within the ensemble. To enable this MPI_COMM_WORLD partitioning, the source code of WRF was modified by substituting MPI_COMM_WORLD with

COMM_model. In case of a single model task, COMM_model would be equal to MPI_COMM_WORLD. In addition to COMM_model, two more communicators are defined for the analysis step in PDAF. COMM_couple facilitates coupling between WRF and PDAF, while COMM_filter, encompasses all processes involved in the initial model task. PDAF provides a template for init_parallel_pdaf, which users can customize as per specific requirements.

**Init_pdaf**: Inserted just before the time-stepping loop in the model code, this routine initializes the PDAF framework. It

specifies parameters for the DA, which may be read from a configuration file or provided via command line inputs. Subsequently, the initialization routine for PDAF is invoked, configuring the PDAF framework and allocating internal arrays, including the ensemble states array. At this juncture, the initial ensemble is initialized. This can be performed using second-





order exact sampling (Pham et al., 1998) from a decomposed covariance matrix. For this, a callback routine, init_ens_pdaf, is called to read the covariance matrix information and generate the initial ensemble. Once PDAF is initialized, information from the initial ensemble is written into the model's field arrays. Subsequently, the initial forecast phase is initialized, entailing a specific number of time steps until the initial analysis step.

**Assimilate_pdaf**: This routine is invoked at the conclusion of each model time step. It calls a filter-specific PDAF routine responsible for computing the analysis step of the selected filter method. Before executing the analysis step, the PDAF routine verifies whether all time steps of a forecast phase have been computed. The analysis step includes additional operations such as handling observations, further described below.

### 3.3 Interfaces for model fields and observation

PDAF Interfaces play a pivotal role in executing model- and observation-specific operations, designed to maintain low complexity. Two types of interfaces are introduced: those for model fields and those for observations.

### 3.3.1 Interface for model fields

This interface encompasses two key routines: collect_state_pdaf and distribute_state_pdaf. These routines are invoked before and after the analysis step, respectively, to facilitate the exchange of information between the WRF model fields and the state vector of PDAF. The routine collect_state_pdaf transfers data from the model fields to the state vector, while distribute_state_pdaf initializes the model fields based on the state vector. Both routines execute across all processes involved in model integrations, each operating within its specific process subdomain. The variables of WRF essential for PDAF are the wind components ($u, v, w$, m/s), perturbation geopotential ($ph$, m2/s), perturbation potential temperature ($th$, K), water vapor mixing ratio ($qv$, kg/kg), cloud water mixing ratio ($qc$, kg/kg), rain water mixing ratio ($qr$, kg/kg), ice mixing ratio ($qi$, kg/kg), snow mixing ratio ($qs$, kg/kg), graupel mixing ratio ($qg$, kg/kg), perturbation pressure ($p$, Pa), density ($rho$, Kg/m3) and base-state geopotential ($phb$, m2/s). Note that some of these variables, namely $p$, $rho$ and $phb$ are exclusively used by the observation operators and remain unaltered by PDAF. Consequently, only the remaining variables are updated and written back to WRF.

Additionally, there is a routine called prepoststep_pdaf that permits users to access the ensemble both before and after the analysis step. This functionality enables pre- and post-processing tasks, such as calculating the ensemble mean, which can be saved to a file. Users can also perform consistency checks, ensuring that variables like hydrological properties remain physically meaningful, and make necessary corrections to state variables if required.

In cases when the analysis step incorporates localization, which is typically the case in high-dimensional models like WRF, additional routines are invoked to handle the localization of the state vector. Initially, these routines ascertain the coordinates and dimension of the local state vector for a given index within a local analysis domain. In WRF-PDAF the local analysis domain is chosen to be a single grid point, in contrast to the implementation in AWI-CM-PDAF (Nerger et al., 2020), which utilizes a vertical column of the model grid as the local analysis domain. Since here the local analysis domain is a single grid





point, the dimension of the local state vector is the number of model fields included in the state vector. The other localization functionality is the initialization of a local state vector from the global state vector according to the index of the local analysis domain. Analogously, the global state vector has to be updated from the local state vector after this has been updated by the local analysis.

### 3.3.2 Interface for observations - Observation Module Infrastructure (OMI)

The implementation utilizes the Observation Module Infrastructure (OMI), a recent extension of PDAF. OMI offers a modular approach to handling observations. In comparison to the traditional approach of incorporating observations with PDAF, OMI presents two notable advantages. Firstly, simplified implementation: OMI considerably reduces the coding effort required to support various observation types, their respective observation operators, and localization. By defining standards how to initialize observation information, OMI simplifies the process and minimizes the coding complexities associated with handling

observations. With this, several routines that had to be coded by the user in the traditional approach are now handled internally by OMI. Secondly, enhanced flexibility: OMI enhances flexibility by encapsulating information about each observation type. This encapsulation prevents interference between different observation types. From the code structure, OMI is motivated by object-oriented programming, but for the sake of simplicity, the actual abstraction of object-oriented code is avoided.

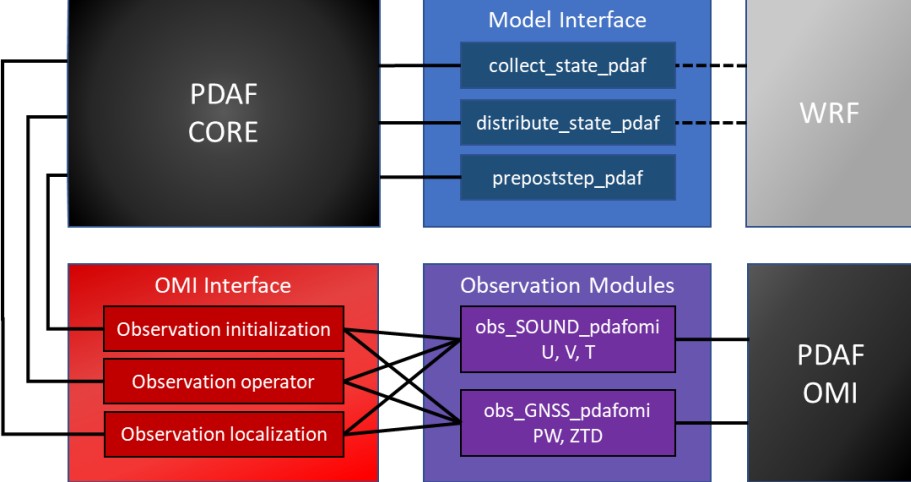

**Figure 2: Sketch of interfaces for model fields and observation (OMI) in the analysis step. The PDAF-core invokes the model interface routines and the OMI interface. The routines collect_state_pdaf and distribute_state_pdaf incorporate information on the model fields from WRF.**

Figure 2 provides a visual representation of these interfaces. The interfaces in the OMI framework encompass three key

components:

**Observation initialization**: For each observation type, a dedicated routine reads observations of that type from a file. Then, it tallies the valid observations, accounting for factors like observation quality flags. The routine initializes the observation




coordinates and observation errors. Additionally, it determines which elements of the state vector are required for the observation operator to compute the model counterpart to an observation. If interpolation is involved in the observation

operator, interpolation coefficients may be calculated. Once these quantities are initialized, an OMI routine is called, transferring the observation information to OMI for use in the PDAF analysis step. In the twin experiments, observation initialization can generate and read synthetic observations.

**Observation operator**: This routine, as described in Section 2.2, implements the observation operator. It takes an ensemble state vector as input and returns the corresponding observed state vector. This operation is performed for each state vector

within the ensemble. The information specifying which elements of the state vector are used in the observation operator and any applicable interpolation weights was initialized by the observation initialization routine. OMI provides some universal operators for interpolations in one, two and three dimensions, including support for triangular grids. The observation operator with interpolation is generic. One just needs to determine interpolation weights and the indices of the elements in the state vector which are combined. For instance, for profile data, the operator for T should be implemented in obs_SOUND_pdafomi.

Furthermore, for complicated remote sensing observation operators, some possible additions would also be implemented. Currently, two operator modules have been implemented, covering sounding observation operators, including U, V, and T, as well as Global Navigation Satellite System (GNSS) observation operators, including Precipitable Water (PW) and Zenith Total Delay (ZTD). These operators are detailed and utilized as described by Shao and Nerger (2023).

**Observation Localization**: The localized analysis, described in Sect. 2.2, necessitates to determine the observations within a

specified distance around a local analysis domain. For each observation type, a dedicated routine calls an OMI routine to identify these observations by calculating the distance between the local analysis domain and observations based on their coordinates. The OMI-routine takes as input a localization radius and the coordinates of the local analysis domain. Internally, OMI performs distance-dependent weighting of observations based on their coordinates. In WRF-PDAF, the local analysis domain consists of a single grid point, hence the observation localization operates in three dimensions, requiring both

horizontal and vertical localization radii to be specified. This contrasts with the observation localization in two dimensions and the use of only horizontal localization radius in AWI-CM-PDAF (Nerger et al., 2020).

This structured approach to model fields and observations, as facilitated by PDAF and OMI, ensures a robust and versatile framework for data assimilation within WRF and other geoscientific models. PDAF provides a model-agnostic framework to create an efficient data assimilation system as well as filter and smoother algorithms. As such, it ensures a clear separation of

concerns between model development, observations, and assimilation algorithms.

## 4 Evaluation of WRF-PDAF

In this section, we delve into the application of WRF-PDAF, specifically focusing on its utility in DA. We particularly aim for evaluating both the parallel performance and the behavior of DA of T profiles.



## 4.1 Compute performance

For evaluation of the performance of WRF-PDAF we use an ensemble of 40 tasks, in which each single WRF task is distributed across a total of 64 processes. As a result, we utilize a grand total of 2560 processes for the ensemble DA. T profiles are placed at 10-grid-point intervals in both the x and y directions. In Figure 3(a), we provide a chart outlining execution times of the various steps of the assimilation procedure.

The breakdown of the key execution times for different phases of the assimilation process over the full experiment is as follows:

The major execution time of roughly 288.6 seconds is needed for conducting the ensemble forecasts over 85 hours is. This time requirement is followed by the communication related to DA coupling (within the communicator COMM_couple), encompassing both data collection and distribution within the ensemble, which takes approximately 27.4 seconds. The initialization stage (in init_ens_pdaf), which involves generating the ensemble, consumes approximately 18.9 seconds. Less time is spend for the DA analysis, involving 30 cycles in the full experiment, which has a total execution time of 7.4 seconds.

Activities associated with DA pre- and post-processing (in prepoststep_pdaf) occupy a combined execution time of 6.6 seconds. The time for the DA analysis can be further broken down into three components: PDAF-Internal Operations, observation handling, and variable transformation. Here, the observation handling require the most time with around 4.9 seconds. The PDAF-internal operations of the LESTKF, like the singular value decomposition and the multiplication of the forecast ensemble with the weight matrix and vector ESTKF demand approximately 1.9 seconds. Variable Transformation: An

additional 0.6 seconds is dedicated to the transformation of variables between the global and local domains. Overall, the execution time for the entire assimilation process, amounting to 349 seconds, is largely dominated by the time required for computing forecasts. For individual cycles, the execution times are distributed as follows: 3.4 seconds for forecasts, 0.9 seconds for coupling communication, 0.2 seconds for DA analysis, and 0.2 seconds for pre/post operations, as demonstrated in Figure 3(b). It is crucial to acknowledge that the execution times can vary depending on the distribution of the program across the

computing resources. Nevertheless, repeated experiments have consistently shown that the timings depicted in Figure 3(b) are representative of the typical performance.

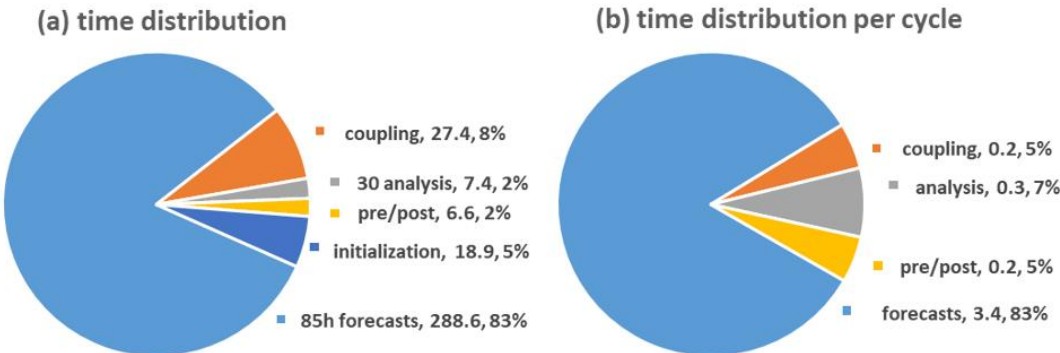





**Figure 3: Flow chart and execution times for different Phases of the DA experiment (Unit: s).**


The numerical experiments, with hourly assimilation of T profiles into WRF, exhibit high efficiency. This efficiency is underscored by an overhead of only up to 20.9% in computing time when compared to the model without assimilation functionality, with an ensemble size of 40. This favorable outcome is largely attributed to the optimization of the ensemble DA program, which prioritizes efficient ensemble integrations between observations, thus reducing the need for disk operations.

Instead, ensemble information is retained in memory and efficiently exchanged through parallel communication during program runtime. The execution time of the DA analysis is influenced by the number of assimilated observations and will increase if more observations are assimilated.

It is important to highlight that the forecasts presented here are derived from an idealized case, characterized by numerous simplifications. For instance, radiation schemes have been omitted in this idealized scenario, resulting in shorter simulation

runtimes compared to real cases. In actual operational scenarios, the model physics would likely be more complex and forecasting times would be notably longer than those in the idealized case. Consequently, when evaluating efficiency by dividing the time dedicated to analysis by that of the forecast, the efficiency values tend to become even more favourable in favour of the assimilation process.

## 4.2 Assimilation results

To assess the influence of the T profiles, they are assimilated here at all vertical columns. Figure 4 presents the root-mean-square error (RMSE) over time and the time-averaged vertical RMSE profiles for T and the two horizontal velocity fields U and V. The primary focus of the experiments is to evaluate the impact of the horizontal localization radius. Notably, the RMSE of the ensemble forecast (ENS) is lower than that of the single control run (CTRL) and the true state (True). This suggests that the ensemble approach itself improves the accuracy of the model prediction. Furthermore, when assimilating T data, the RMSE

of T (Fig. 3) is much lower than that of ENS during the analysis period. Thus, the assimilation process significantly enhances the accuracy of the model prediction. Among the experiments ANA3, ANA4, and ANA5, there are similarities in RMSE values, with ANA4 exhibiting the lowest RMSE among all the experiments in Table 1. Smaller and large localization radii lead to increased RMSE. When the assimilation is stopped, the RMSE value increase significantly. At the end of the experiment after 34 hours of free forecast, the RMSE for T from the ANA-experiments is at a similar level as the RMSE of ENS.

With the aid of flow-dependent cross-variable background error covariances, the multivariate assimilation of T profiles not only reduces the errors of the T field but also leads to improvements in the U and V fields. Specifically, in Fig. 3, the RMSEs for U from the experiments ANA4, ANA5 and ANA6 appear quite similar, with ANA5 exhibiting the lowest RMSE among all the experiments. In Figure 4, a similar pattern is observed for V, with ANA4 having the lowest RMSE among all the experiments. This demonstrates that the assimilation of T data contributes to reduced forecast errors and more consistent

forecasts. Note that the localization radius notably influences the assimilation result. An appropriately chosen localization radius leads to improvements in the background model. However, when the localization radius is set to 0, the RMSEs of U



and V from ANA0 become higher than those from ENS during the forecast period, as shown in Figures 3b and 3c. Additionally, after the final assimilation cycle at 051400, the RMSE of T from ANA0 sharply increases at the first forecast step, 051500, as visible in Figure 4. These special behaviors are due to the phenomenon of overfitting, i.e., the model is adjusted not only to

the data but also to the noise (Nerger et al., 2006). In contrast the cases ANA1 to ANA10 show a lower RMSE for U and V at the end of the experiment. Thus, the assimilation improves the velocity field and a part fo the improvement remains present in the ensemble also after 34 hours of free forecast.

In the time-averaged RMSE profiles, improvements induced by the assimilation are visible in all levels of the model. They are lowest at the uppermost layers.

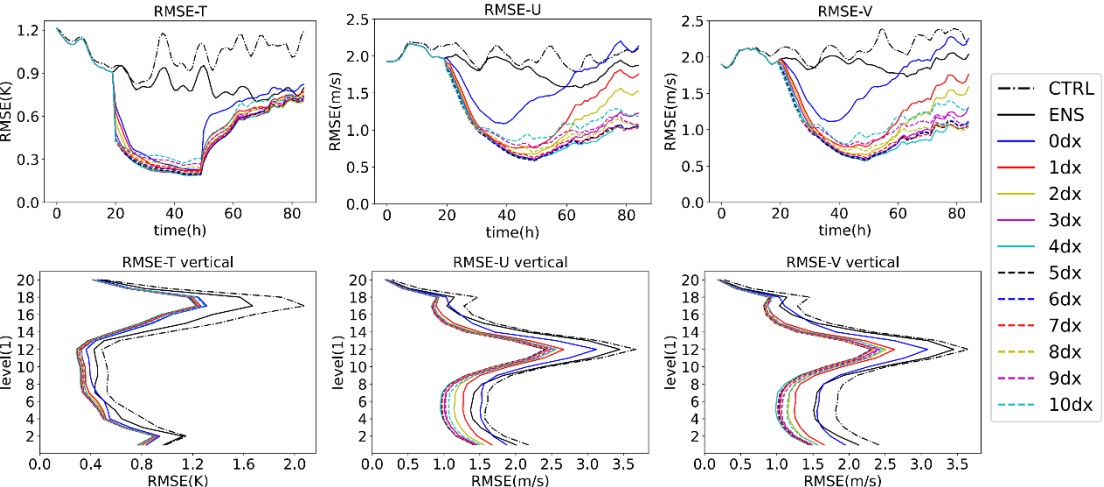


**Figure 4: The RMSEs of T, U and V from 031200 to 070000 (a: RMSE of T in time series; b: RMSE of U; c: RMSE of V in time series; d: vertical average of T RMSE; e: vertical average of U RMSE; f: vertical average of V RMSE).**

Figure 5 illustrates the relationship between the localization radius and the RMSEs of T, U, and V. To achieve the smallest

RMSEs, a localization radius of 4dx is a desirable selection when assimilating the full set of observations. However, short localization radii (< 4dx) are detrimental to balance. Conversely, long localization radii (> 4dx), when compared to the optimal radius, may lead to larger errors and imbalances due to presumed spurious correlations, aligning with findings by Greybush et al. (2011). It is important to emphasize that the experiments conducted based on different localization radii serve as fundamental demonstrations of the functionality of the DA program, with in-depth analysis not being the primary focus of this

study.



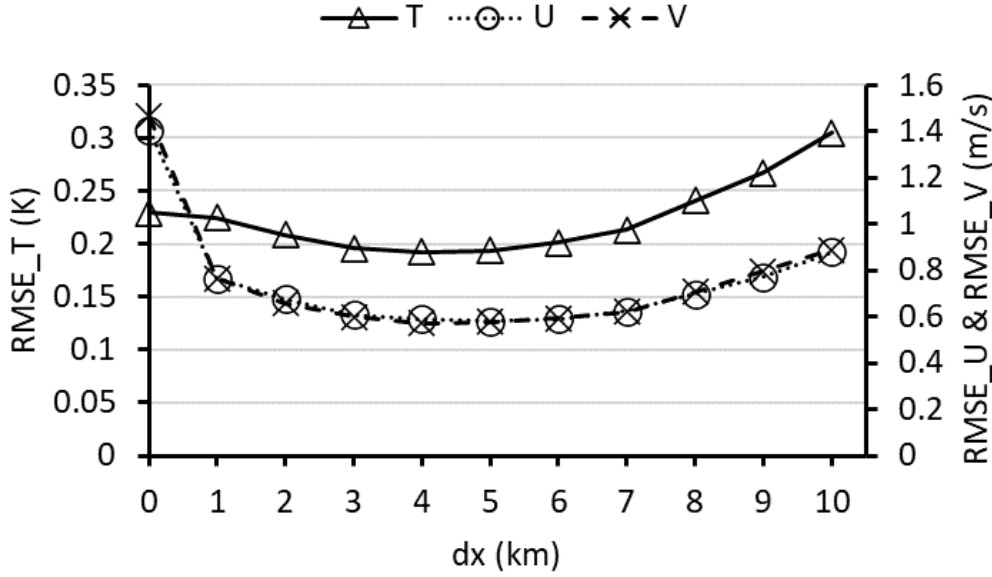

**Figure 5: The RMSEs of T, U, and V using different localization radii.**

## 5 Discussion and conclusions

This paper introduces and evaluates WRF-PDAF, a fully online-coupled ensemble DA system that couples the atmosphere model WRF with the data assimilation framework PDAF. In comparison to AWI-CM-PDAF 1.0 (Nerger et al. 2020), several key distinctions stand out. Firstly, the coupled models diverge significantly. AWI-CM represents a climate model, whereas WRF is an atmospheric regional model. Consequently, the framework, state vector definition, and incorporated observations are fundamentally dissimilar. Secondly, the PDAF version varies. Notably, the introduction of the newly developed OMI has led to a divergence in code structure. This marks the inaugural use of OMI in implementing observation interfaces. Lastly, there are disparities in computational performance and assimilation outcomes. Importantly, this novel endeavour underscores the adaptability of PDAF, as it proves its efficacy not only in large-scale climate system models but also in mesoscale regional atmospheric models.

A key advantage of the WRF-PDAF configuration is its ability to concurrently integrate all ensemble states, eliminating the need for time-consuming distribution and collection of ensembles during the coupling communication. This innovative online DA system eliminates the necessity for frequent model restarts, a common requirement in offline DA systems. Without the need for model restarts and file I/O operations, the extra time required for DA, including the analysis, communication, and pre/post operations, amounts to only 20.6% per cycle in our test assimilation of T profile observations every hour for 30 cycles. Twin experiments focusing on an idealized tropical cyclone configuration were conducted to validate that the WRF-PDAF system works correctly. The results underscore the effectiveness of the WRF-PDAF system in assimilating T profile data,



leading to significant enhancements not only in three-dimensional temperature fields but also in three-dimensional wind components (U and V). The choice of an optimal localization radius is demonstrated, although it is important to note that the localization distance can vary depending on the specific case.

The code structure using interface routines inserted into the WRF model code and observation-specific OMI routines make the assimilation framework highly flexible. Further, the abstraction in the analysis step, which uses only state and observation

vectors without accounting for the physical fields, allows one to separate the development of advanced DA algorithms from the development of the model. Therefore, ensuring a clear separation of concerns becomes imperative, a requirement for the efficient development of intricate model codes and their adaptation to contemporary computing systems (Lawrence et al., 2018).. The separation allows all users with their variety of models to use newly implemented DA methods by updating the PDAF library and, if the new method has additional parameters, to specify these additional DA. To guarantee compatibility

across various library versions, the interfaces to the PDAF routines remain unaltered. The abstraction in the analysis step and the model-agnostic code structure also allows users to apply the assimilation framework independently of the specific research domain.

The example here uses a parallelization in which the analysis step is computed using the first model task and the same domain decomposition as the model. Other parallel configurations are possible. Although fully parallel execution of the assimilation

program is highly efficient, it is constrained by the maximum job size permitted on the computer. The model used in the example here can scale even further than the 64 processes used for WRF. Hence, on the same computer, one could either execute a larger ensemble with fewer processes per model, resulting in a longer runtime, or opt for a smaller ensemble, which would reduce the runtime. The number of processes should be set so that the requirements on the ensemble size for a successful assimilation can be fulfilled. The other aspect is the required memory. The analysis step needs the whole ensemble stored in a

domain-decomposed way. Thus, the complete ensemble is collected on the processes of task 1, which calculate the analysis step. In extreme cases this might overload the available memory. For larger applications, one might need to obtain a compute allocation at larger computing sites, such as national compute centres.

In conclusion, this study elucidates the DA program by enhancing the WRF model code and employing in-memory data transfers between the model and PDAF. The Observation Module Infrastructure (OMI) plays a pivotal role in handling

observational data, encompassing observation initialization, observation operators, and observation localization. While the current implementation includes operators for profile data (T, U, and V) and GNSS data (PW and ZTD), it maintains flexibility for incorporating complex remote sensing observation operators. The exemplary outcomes of perfect twin experiments affirm the effectiveness of the WRF-PDAF system in assimilating observations. Importantly, given that real-world forecasting times may be longer than ideal case scenarios operational DA performance could be even more efficient. Overall, the online WRF-

PDAF system provides an efficient and promising framework for implementing high-resolution mesoscale forecasting and reanalysis, bridging the gap between cutting-edge research and practical applications in weather forecasting and climatology.





**Code availability**

Code can be download at https://doi.org/10.5281/zenodo.8367112.

**Data availability**

Dataset can be download at https://doi.org/10.5281/zenodo.10083810.

**Author contribution**

CS and LN planned the campaign; CS performed the experiments, analysed the data and wrote the manuscript draft; LN
reviewed and edited the manuscript.

**Competing interests**

The authors declare that they have no conflict of interest.

**Financial support**

Changliang Shao was supported by the China Scholarship Council for one year research at AWI (No. 202105330044).

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
