# Peer review of "WRF-PDAF v1.0: Implementation and Application of an Online Localized Ensemble Data Assimilation Framework"

_EGUsphere, 2023_

## Referee Comment (RC2)

The manuscript proposed an online parallelized data assimilation framework (PDAF) coupled with WRF-ARW. The online DA framework prevents the huge data I/O from the hard disk, which makes the framework more suitable for the operational application. As the first work of PDAF coupled with WRF-ARW, several experiments are conducted to examine the computational efficiency of PDAF and the performance of LESTKF through T profile assimilation. In general, the manuscript is well-written, but there are some minor issues should be addressed before publication.

Line 118-119. It may be better to state the reason of assimilating T profiles in the introduction.

Please briefly describe the features of LESTKF that distinguish the algorithm from others like LETKF and EnSRF. Or briefly introduce the advantage of LESTKF so that readers do not need to read early papers to understand why the authors selected the algorithm.

For equations (6) - (7), could the authors add some words to explain to readers how to deal with the situation when rank deficiency exists?

Line 148. Why the authors named the control state as 'True' and the true state as 'CTRL', which is somewhat confusing. The names are inconsistent with those in Line 373.

Please give the resolution of T profiles in the description of experimental design, rather than Section 4.1. Additionally, is the assumption of observation errors being uncorrelated still valid? What is the vertical resolution of a temperature profile? If all model levels are used to generate an observation profile, the above assumption may not be valid.

Is it possible for the authors to add a schematic plot describing how the ensemble runs and how the online PDAF obtains data from the ensemble?

Figure 1. What does the n in "do i=1,n" mean? Does n repsent the total number of time steps?

How OMI deals with QC? For instance, if the OMB (observation minus background) of an observation is larger than 5 times the standard deviation, the observation should be discarded. Does the OMI interface in the manuscript use an individual member or the ensemble mean as the background? It is not an issue for OSSE but an issue for real data assimilation

How to deal with the localization of satellite observations?

Line 381: Is it Figure 4?

Line 387: Also Figure 4 not Figure 3?

Figure 4: regarding the vertical average, is it computed at a specific moment or over a time span?

Line 401-402: Whether a long radius is an optimal choice depends on the quality of background error covariance and the length scale of error samples. For the discussion here, please refer to some work discussing the multiscale background error covariance

---

## Author Comment (AC2)

The manuscript proposed an online parallelized data assimilation framework (PDAF) coupled with WRF-ARW. The online DA framework prevents the huge data I/O from the hard disk, which makes the framework more suitable for the operational application. As the first work of PDAF coupled with WRF-ARW, several experiments are conducted to examine the computational efficiency of PDAF and the performance of LESTKF through T profile assimilation. In general, the manuscript is well-written, but there are some minor issues should be addressed before publication.

R: We greatly appreciate the reviewer for the encouraging comments and insightful suggestions listed below. A point-by-point response to the reviewers' comments is provided below. The reviews' comments are in black font and our responses are in blue font.

Line 118-119. It may be better to state the reason of assimilating T profiles in the introduction.

Re: As we know, T is a much common variable, and it is more stable than Qvaper or Humidity, as well as T is easier to understand. Assimilating T profiles is adopted to test the efficiency and performance, especially when the observations are located on all of the model grid points. A description has been added there.

Please briefly describe the features of LESTKF that distinguish the algorithm from others like LETKF and EnSRF. Or briefly introduce the advantage of LESTKF so that

readers do not need to read early papers to understand why the authors selected the algorithm.

Re: A description has been added to the last paragraph as 'Like the LETKF, the LESTKF performs a sequence of local analyses, but it does the calculations directly in the error subspaces spanned by the ensemble. This can lead to computational savings compared to the LETKF."

A comparison to the EnSRF does not look relevant, since this filter does not appear at all in the manuscript.

For equations (6) - (7), could the authors add some words to explain to readers how to deal with the situation when rank deficiency exists?

Re: The matrix $A^{-1}$ is typically never rank deficient since $R^{-1}$ is full rank. Anyway, if $A^{-1}$ is rank deficient, the calculations in Eqns. (6) and (7) can be performed only for the none-zero eigenvalues. We add this information after Eqn (8) in the manuscript.

Line 148. Why the authors named the control state as 'True' and the true state as 'CTRL', which is somewhat confusing. The names are inconsistent with those in Line 373.

Re: Corrected in the revision.

Please give the resolution of T profiles in the description of experimental design, rather than Section 4.1. Additionally, is the assumption of observation errors being uncorrelated still valid? What is the vertical resolution of a temperature profile? If all model levels are used to generate an observation profile, the above assumption may not be valid.

Re: The observations are located on all of the model grid points, so the T profiles have the same resolution with the model grid.

Actually, all model levels are indeed used to generate an observation profile. However, the random Gaussian error is also added to the model value on these model levels. To this end, the assumption of observation errors being uncorrelated is still valid.

Is it possible for the authors to add a schematic plot describing how the ensemble runs and how the online PDAF obtains data from the ensemble?

Re: We have plotted a diagram in the manuscript to describe this process. The information was added in line 435 and line 445.

[Figure]

Figure 1. What does the n in "do i=1,n" mean? Does n repsent the total number of time steps?

Re: Yes, as you said, n represents the total number of time steps. We clarify this in the title of Fig. 1.

How OMI deals with QC? For instance, if the OMB (observation minus background) of an observation is larger than 5 times the standard deviation, the observation should be discarded. Does the OMI interface in the manuscript use an individual member or the ensemble mean as the background? It is not an issue for OSSE but an issue for real data assimilation

Re: QC is usually done in the observation module on the user side. However, the latest release of PDAF (V2.2) allows the user to specify the exclusion factor (5 in your example) and computes the OMB with regard to the ensemble mean. We add to the manuscript in line 302 'OMI also provide some features relating to quality control. E.g. an observation can be excluded if its value deviates too much from the ensemble mean'.

How to deal with the localization of satellite observations?

Re: For satellite observations, the relevant coordinates used for distance calculation are defined by the user. We clarified this in line 333.

Line 381: Is it Figure 4?

Re: Corrected in the revision.

Line 387: Also Figure 4 not Figure 3?

Re: Corrected in the revision.

Figure 4: regarding the vertical average, is it computed at a specific moment or over a time span?

Re: The vertical average is computed over the whole simulation time, which is from the start time to the end time. Description is added in the manuscript in line 384.

Line 401-402: Whether a long radius is an optimal choice depends on the quality of background error covariance and the length scale of error samples. For the discussion here, please refer to some work discussing the multiscale background error covariance

Re: 'For reference, Wang and Liu (2019) discussed the relationship between the observation radius and the background error covariance.' We have added this information in line

Wang, S., and Liu, Z. (2019). A radar reflectivity operator with ice-phase hydrometeors for variational data assimilation (version 1.0) and its evaluation with real radar data. Geoscientific Model Development, 12(9), 4031-4051.

---

## Author Response (AR2)

**#Author's response to reviewer1:**

The authors implemented an online data assimilation framework in the WRF-ARW weather forecasting system, and evaluated its performance utilizing a twin experiment setup of an ideal tropical cyclone case. The results indicate that this assimilation framework is effective in improving the prediction skill of the model and is computationally efficient. It would be really useful to the community if such performance can be reproduced in real-world cases, and I suggest the authors to showcase a successful real-world application to further enhance the manuscript. Anyway, I believe this development work is interesting to the community and should be tested in more real-world cases by the community.

The text is overall well-crafted and easily comprehensible, and the figures effectively convey information, although there is room for improvement in their presentation quality. I have only a few minor comments listed below. Once those have been addressed, I recommend the work be accepted for publication.

Reply: We greatly appreciate your very valuable comments. These are very helpful in guiding the improvement of our manuscript. A point-by-point response to the comments is provided below. We hope this could address the issues. The reviews' comments are in black font and our responses are in blue font.

- L148-149: It seems that the labels for Exp. 1 and Exp. 2 are mistakenly swapped. In addition, it's not clear what the CTL case is. My guess is that it's a single member free

run without data assimilation, but the authors may consider adding some description for each case as well as the idea of the twin-experiment setup so that readers unfamiliar with the experiment can understand.

Reply: Thanks for your comments. Now, the two labels have been swapped back to the correct positions. Moreover, to make the definition clear, descriptions of each case and basic idea of the twin-experiment setup have been added in line 150-152 as "The CTRL is derived from a single free run, which mirrors the True in all aspects, except for a 60-hour delay in its starting time. The ENS, comprising 40 ensemble members, is generated by introducing an initial perturbation to the CTRL. According to the ENS, the DA twin experiments are implemented by assimilating observations using different localization radii."

- "3.1 Subsection (as Heading 2)" seems like unfinished writing.

Reply: Thanks for your comments. "3.1 Subsection (as Heading 2)" has been corrected to "Description of PDAF"

- L185: "in in"

Reply: Thanks for your comments. "in in" has been corrected to "in"

- The teletype (monospace) font should be applied for routine names.

Reply: Thanks for your comments. We rechecked the content, and the font of all routine names has been changed to 'consolas', including the main text and figures.

- L373: How can the RMSE of ENS be lower than that of the true state? In addition, the expression "single control run" seems odd. Perhaps the authors mean "single member control run"?

Reply: Thanks for your comments. In line 373, "and the true state" has been changed to "compared to the true state". Furthermore,    the odd expression "single control run" has been changed to "control run" to make the context consistent. As the currently added description in line 150-152, the control run is defined as a single run, actually the same meaning with "single member control run".

- L377: "Smaller and large" -> "Smaller and larger" ?

Reply: Thanks for your comments.    In order to avoid mistake and ambiguity, "Smaller and large" has been changed to "Compared to ANA4, either smaller or larger"

- L381: "in Fig. 3" --> "in Fig. 4" ?

Reply: Thanks for your comments. Corrected.

- L387: "Figure 3b and 3c" --> "Figure 4b and 4c" ?

Reply: Thanks for your comments. Corrected. in addition, the label (a),(b),...,(f) have been added to the subfigures.

- L400: Can we expect that the selection of 4dx is a good choice for realistic cases? Or is the optimal selection case by case?

Reply: Thanks for your comments. Actually, the optimal selection is case by case. As we noted in line 405-407, "It is important to emphasize that the experiments conducted based on different localization radii serve as fundamental demonstrations of the functionality of the DA program, with in-depth analysis not being the primary focus of this study." Even in the same case, different assimilation observation (i.e. U or V) may have different optimal selection. To avoid misleading, we added a clarification in line 407 as "In addition, the optimal selection is case dependent."

**#Author's response to reviewer2:**

The manuscript proposed an online parallelized data assimilation framework (PDAF) coupled with WRF-ARW. The online DA framework prevents the huge data I/O from the hard disk, which makes the framework more suitable for the operational application. As the first work of PDAF coupled with WRF-ARW, several experiments are conducted to examine the computational efficiency of PDAF and the performance of LESTKF through T profile assimilation. In general, the manuscript is well-written, but there are some minor issues should be addressed before publication.

R: We greatly appreciate the reviewer for the encouraging comments and insightful suggestions listed below. A point-by-point response to the reviewers' comments is provided below. The reviews' comments are in black font and our responses are in blue font.

Line 118-119. It may be better to state the reason of assimilating T profiles in the introduction.

Re: As we know, T is a much common variable, and it is more stable than Qvaper or Humidity, as well as T is easier to understand. Assimilating T profiles is adopted to test the efficiency and performance, especially when the observations are located on all of the model grid points. A description has been added there.

Please briefly describe the features of LESTKF that distinguish the algorithm from others like LETKF and EnSRF. Or briefly introduce the advantage of LESTKF so that readers do not need to read early papers to understand why the authors selected the algorithm.

Re: A description has been added to the last paragraph as 'Like the LETKF, the LESTKF performs a sequence of local analyses, but it does the calculations directly in the error subspaces spanned by the ensemble. This can lead to computational savings compared to the LETKF."

A comparison to the EnSRF does not look relevant, since this filter does not appear at all in the manuscript.

For equations (6) - (7), could the authors add some words to explain to readers how to deal with the situation when rank deficiency exists?

Re: The matrix $A^{-1}$ is typically never rank deficient since $R^{-1}$ is full rank. Anyway, if $A^{-1}$ is rank deficient, the calculations in Eqns. (6) and (7) can be performed only for the none-zero eigenvalues. We add this information after Eqn (8) in the manuscript.

Line 148. Why the authors named the control state as 'True' and the true state as 'CTRL', which is somewhat confusing. The names are inconsistent with those in Line 373.

Re: Corrected in the revision.

Please give the resolution of T profiles in the description of experimental design, rather than Section 4.1. Additionally, is the assumption of observation errors being uncorrelated still valid? What is the vertical resolution of a temperature profile? If all model levels are used to generate an observation profile, the above assumption may not be valid.

Re: The observations are located on all of the model grid points, so the T profiles have the same resolution with the model grid.

Actually, all model levels are indeed used to generate an observation profile. However, the random Gaussian error is also added to the model value on these model levels. To this end, the assumption of observation errors being uncorrelated is still valid.

Is it possible for the authors to add a schematic plot describing how the ensemble runs and how the online PDAF obtains data from the ensemble?

Re: We have plotted a diagram in the manuscript to describe this process. The information was added in line 435 and line 445.

[Figure]

Figure 1. What does the n in "do i=1,n" mean? Does n repsent the total number of time steps?

Re: Yes, as you said, n represents the total number of time steps. We clarify this in the title of Fig. 1.

How OMI deals with QC? For instance, if the OMB (observation minus background) of an observation is larger than 5 times the standard deviation, the observation should be discarded. Does the OMI interface in the manuscript use an individual member or the ensemble mean as the background? It is not an issue for OSSE but an issue for real data assimilation

Re: QC is usually done in the observation module on the user side. However, the latest release of PDAF (V2.2) allows the user to specify the exclusion factor (5 in your example) and computes the OMB with regard to the ensemble mean. We add to the manuscript in line 302 'OMI also provide some features relating to quality control. E.g. an observation can be excluded if its value deviates too much from the ensemble mean'.

How to deal with the localization of satellite observations?

Re: For satellite observations, the relevant coordinates used for distance calculation are defined by the user. We clarified this in line 333.

Line 381: Is it Figure 4?

Re: Corrected in the revision.

Line 387: Also Figure 4 not Figure 3?

Re: Corrected in the revision.

Figure 4: regarding the vertical average, is it computed at a specific moment or over a time span?

Re: The vertical average is computed over the whole simulation time, which is from the start time to the end time. Description is added in the manuscript in line 384.

Line 401-402: Whether a long radius is an optimal choice depends on the quality of background error covariance and the length scale of error samples. For the discussion here, please refer to some work discussing the multiscale background error covariance

Re: 'For reference, Wang and Liu (2019) discussed the relationship between the observation radius and the background error covariance.' We have added this information in line

Wang, S., and Liu, Z. (2019). A radar reflectivity operator with ice-phase hydrometeors for variational data assimilation (version 1.0) and its evaluation with real radar data. Geoscientific Model Development, 12(9), 4031-4051.